# MultiWave: Multiresolution Deep Architectures through Wavelet Decomposition for Multivariate Timeseries Forecasting and Prediction

## Abstract

One of the challenges in multivariate time series modeling is that changes in signals occur with different frequencies, even when the sampling rate is consistent across signals. In the case of multivariate time series prediction, the outcome is also determined by patterns of different frequencies. These encapsulate both long-term and short-term effects, which have so far not been sufficiently leveraged by deep learning time series models. We fill this gap by introducing a framework, called MultiWave, which augments any deep learning time series model with components operating at the intrinsic frequencies of the signals. MultiWave applies wavelet decomposition on each signal to obtain subsignals of different frequencies and groups all subsignals in the same frequency band together to train a component. The output of the components is combined through a gating mechanism that removes irrelevant frequencies for the given predictive task. We show that MultiWave accurately determines the informative frequency bands and that the augmented models including components trained to operate on those bands outperform the original models. We further show that applying MultiWave on top of different deep learning models improves their performance in several real-world applications.

## 1 Introduction

Multivariate time series prediction has long been a crucial task in machine learning, as it has important applications in many fields such as healthcare, traffic flow, and economic forecasting. However, the final prediction in these applications can depend on many factors, such as information at different frequencies, long-term and short-term changes in input signals. Moreover, in many tasks, observations come from multiple sources and are often collected at various sampling rates. Here, we propose a model-agnostic approach that can leverage temporal dependencies at different frequencies and scales in multivariate time series data that might be collected with multiple sampling rates (multirate time series data) using multilevel discrete wavelet decomposition.

There are two important categories of methods for time series analysis: Time-domain methods that consider the time series as a sequence of ordered points in time and frequency-domain methods that use transform algorithms such as Fourier transform and Z-transform to analyze the original sequence in the frequency spectrum. Deep learning-based methods that are introduced into time series analysis, such as recurrent neural networks (Williams & Zipser, 1989), Convolutional Neural Networks (CNN) (Zheng et al., 2016) and more recently transformers (Wen et al., 2022) achieve state-of-the-art results in many applications (Lai et al., 2022; Tipirneni & Reddy, 2021; Huang et al., 2022). However, they have two notable shortcomings in handling multivariate time series data. First, most of these methods only use information available in the time domain and cannot leverage information present in the frequency domain of the signals. Additionally, these methods cannot directly model signals that are collected with different frequencies (multirate signals), and upsampling or downsampling these time series to a single rate can artificially introduce or remove some important temporal dependencies Che et al. (2018a); Tipirneni & Reddy (2021); Che et al. (2018b).

To overcome these deficiencies, we propose a novel model-agnostic framework, which uses discrete wavelet decomposition to break signals into different frequency components (*subsignals*), group

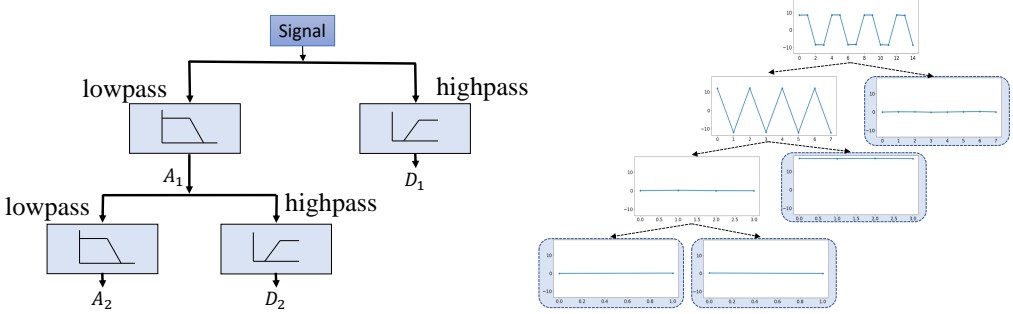

Figure 1: Multi level Discrete Wavelet Decomposition, image on the left shows how lowpass and highpass filters are used to decompose signals in multi level discrete wavelet decomposition and the image on the right shows a signal getting decomposed by Haar wavelet. As you can see the resulted signals are zero except the signal matching the true frequency of the original signal.

subsignals with similar frequencies into separate time series models, and then combine the output of the models to make a prediction. This framework brings the following improvements to multivariate time series modeling: 1) Model-agnostic, MultiWave can be applied to any neural network-based time series model. 2) Uses the information available in both the time and frequency domains. 3) Reduces the amount of variation between the sampling frequency of multiresolution signals that are modeled together. 4) Provides unique insight into which frequencies of the signals are important for a given task.

## 2 BACKGROUND

### 2.1 NOTATION

We denote multivariate and multirate time series data with $m$ signals collected before time $T$ as a set of signals $X^{1:T} = \{x_1^{1:T}, x_2^{1:T}, ..., x_m^{1:T}\}$ where each signal is collected at initial rates $R = \{r_1, r_2, ..., r_m\}$. The length of each signal is proportional to its collected rate $Len_i = \frac{T}{r_i}$. The problem is given $X^{1:T}$, we want to predict a label $y$, which can be continuous (regression) or discrete (classification). In the rest of the paper, we will remove the time indication for the signals and show the set of signals as $X$ and the signal $i$ as $x_i$. We show the sampling rate of a signal $x$ as $f_s(x)$.

### 2.2 MULTILEVEL DISCRETE WAVELET DECOMPOSITION

We use wavelet decomposition to break down the signals into different frequencies. Wavelet decompositions (Daubechies, 1992) are well-known methods for capturing information in time series in both the time and frequency domains. They have been used successfully as a preprocessing step for neural networks (Liu et al., 2013; Wang et al., 2020a) and as an integral part of them (Subasi et al., 2006; Zhang et al., 1995; Wang et al., 2018; Kumar et al., 2021). Multilevel discrete wavelet decomposition can extract multilevel time-frequency features from a time series by iteratively applying low-pass and high-pass filters derived from wavelets to the signal. The formula for this decomposition is given below:

$$x(t) \approx \sum_k A_{L,k}\phi_{L,k}(t) + \sum_k D_{L,k}\Psi_{L,k}(t) +$$
$$\sum_k D_{L-1,k}\Psi_{L-1,k}(t) + ... + \sum_k D_{1,k}\Psi_{1,k}(t)$$

$\Psi_{s,\tau}$ is the mother wavelet with scale $s$ and time $\tau$ and $\phi$ is the father wavelet. This multilevel wavelet decomposition converts the input signal $x(t)$ into signals $A_L = \bigcup_k A_{L,k}$, which is a coarse general approximation of the signal (low frequency) and the detail coefficients $D_L = \bigcup_k D_{L,k}, D_{L-1} = \bigcup_k D_{L-1,k}, ..., D_1 = \bigcup_k D_{1,k}$ that influence the function on various scales. Figure 1 depicts this decomposition. To simplify the notation, we show the decomposition of a signal $x$ as a set

$S(x) = \{D_1, D_2, \ldots, D_L, A_L\}$ that includes signals retrieved when decomposing $x(t)$ at $L$ levels. This notation allows us to denote $D_1$ as $S_1(x)$, $D_2$ as $S_2(x)$, ..., $D_L$ as $S_L(x)$, and $A_L$ as $S_{L+1}(x)$.

There are many different wavelets introduced in the literature, such as Haar, Daubechies, and Biorthogonal (Haar, 1909; Cohen et al., 1992; Daubechies, 1992) that can be used here. Our framework is independent of the type of wavelet used.

## 2.3 RELATED WORK

**Multirate time series classification** Deep learning-based methods that are used for time series analysis, such as recurrent neural networks (Williams & Zipser, 1989), Convolutional Neural Networks (CNN) (Zheng et al., 2016) and more recently transformers (Wen et al., 2022) achieve state-of-the-art results in many applications (Lai et al., 2022; Huang et al., 2022). However, these approaches cannot be used directly for multirate time series data. To be able to apply these models in these settings, signals should be aligned by upsampling lower frequency signals or downsampling higher frequency signals, leading to errors in prediction and loss of information Che et al. (2018a); Tipirneni & Reddy (2021). These models fall short in two important ways when handling multivariate time series data. First, most of these methods only use information available in the time domain and are not designed to take advantage of the information present in the frequency domain of signals. Furthermore, these methods cannot be easily used on signals that are collected with different frequencies (multiresolution signals). To be able to apply these models, the signals should be aligned either by upsampling the lower frequency signals, or downsampling the higher frequency signals that lead to errors in prediction and loss of information Che et al. (2018a); Tipirneni & Reddy (2021). There are several models proposed to solve this problem, such as Che et al. (2018a) and Tipirneni & Reddy (2021) which can inherently model irregularly sampled time series data and thus can model multirate time series data without aligning the signals. There are other methods, such as Che et al. (2018b); Armesto et al. (2008); Safari et al. (2014) that use architectures specifically developed for multirate time series data. All these approaches only consider the information available in the time domain of the time series data, while our model is specifically developed for multirate data and is able to leverage the information in the frequency domain.

**Frequency analysis of time series** Frequency analysis of time series is an extensively studied subject in the signal processing community. Methods such as the discrete Fourier transform (Bracewell & Bracewell, 1986), the discrete wavelet transform (Daubechies, 1992), and Z-Transform (Foster, 1996) have been used to analyze time series. For deep learning models, similar methods have been used in the preprocessing steps (Cui et al., 2016; Yuan et al., 2017; Song et al., 2021) or as part of neural networks (Koutnik et al., 2014; Lee-Thorp et al., 2021). Most of these models focus on univariate time series data and cannot be directly used on multivariate and multirate time series data.

**Wavelet decomposition** Wavelet decompositions (Daubechies, 1992) are well-known methods for capturing information in time series in both the time and frequency domains. They have been used successfully as a preprocessing step for neural networks (Liu et al., 2013; Wang et al., 2020a; Alhnaity et al., 2021; Kim et al., 2021; Althelaya et al., 2021; Zucatelli et al., 2021) and as an integral part of them (Subasi et al., 2006; Zhang et al., 1995; Wang et al., 2018; Guo et al., 2022; Li et al., 2021). Wang et al. (2018) proposes the methodology closest to our method, implementing a trainable wavelet decomposition framework, which can be trained with the rest of the network. Although similar to our method, this paper uses wavelet decomposition to extract frequency-based information from datasets and modeling them using different model components; it cannot be applied to multivariate and multirate time series data and does not use feature masks to remove useless frequency components of the signals from the framework. Furthermore, they propose a model to be used, while our architecture is model-agnostic and can be applied to any time series model.

## 3 MULTIWAVE

### 3.1 SIGNAL DECOMPOSITION AND FREQUENCY GROUPING

Figure 3 shows the overall structure of the framework for two signals when the signals $x_1$ and $x_2$ are collected with frequencies of $64Hz$ and $32Hz$, respectively. We will use discrete wavelet decomposition to decompose each signal into different frequency groups, so for $m$ input signals, $X = \{x_1, x_2, \ldots, x_m\}$ we will have a set of decomposed signals, $S(X) = \{S(x_1), S(x_2), \ldots, S(x_m)\}$.

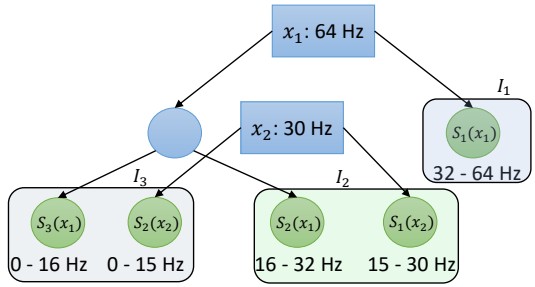

Figure 2: The decomposition and grouping of two signals sampled at 64Hz ($x_1$) and the other one at 30Hz ($x_2$).

In the case where all signals are sampled at the same rate, all elements of $S(X)$ will have the same number of levels and the frequencies at each level will be the same. Therefore, we group all frequency components at the same level into a model component. Therefore, the component $i$, denoted as $\Phi_i$, receives the input $I_i = \{S_i(x_1), \ldots, S_i(x_m)\}$. The number of components is equal to the level of wavelet decomposition $L$. Then we concatenate the outputs of all models and use a fully connected layer to obtain the output.

$$\hat{y} = FC(\Phi_1(I_1) \oplus \Phi_2(I_2) \oplus \cdots \oplus \Phi_L(I_L)) \tag{1}$$

If the signals are collected with different sampling rates, we will group the subsignals with frequencies that are close to each other together into a model component. In this case, the number of model components is equal to the maximum level of signal decomposition $L_{max} = max(\{L_1, L_2, \ldots, L_m\})$ determined by the signal with the highest sampling rate. To better illustrate this, assume, without loss of generality, that we have two signals $m = 2$ where signal $x_2$ is collected twice as often as signal $x_1$, $f_s(x_2) = 2 \times f_s(x_1)$. In this case, $L_2 = L_1 + 1$ since the frequencies are reduced by half at each level of wavelet decomposition, we have:

$$f_s(x_1) = S_1(f_s(x_2)), f_s(S_1(x_1)) = f_s(S_2(x_2)), \ldots, f_s(S_i(x_1)) = f_s(S_{i+1}(x_2))$$

So, to model signals with the same rates in each component, in this case, the inputs of the model would be:

$$I_1 = \{S_1(x_2)\}, I_2 = \{S_1(x_1), S_2(x_2)\}, \ldots, I_{L_2} = \{S_{L_1}(x_1), S_{L_2}(x_2)\}$$

If the rates of signals for one component are not equal, we will over-sample the shorter signal to match the longer signal. The difference between the number of levels between the decomposition of two signals is $\log(\frac{f_s(x_2)}{f_s(x_1)})$, so if this proportion is not a power of 2, the shorter signal at each component level should be oversampled to the closest power of 2 to match the corresponding level of the other signal. For example, if $f_s(x_2) = 6 \times f_s(x_1)$ in the above example, the input will be:

$$I_1 = \{S_1(x_2)\}, I_2 = \{S_2(x_2)\}, I_3 = \{S_3(x_2), S_1(x_1)\} \ldots, I_{L_2} = \{S_{L_1}(x_1), S_{L_2}(x_2)\}$$

since for the component $\Phi_3$, $\frac{f_s(S_3(x_2))}{f_s(S_1(x_1))} = 1.5$, $S_1(x_1)$ should be oversampled by a proportion of 1.5. Note that the oversampling proportion will always be less than 2. Figure 2 shows how this decomposition and grouping would work for two signals with different sampling rates.

Using this approach, the components trained on lower frequencies learn long-term changes in the data, while the faster frequency components learn short-term fast-changing trends in the data. Furthermore, since the signals are grouped with respect to their sampling rates, the signals that are input into each component have similar frequencies, which significantly reduces the amount of oversampling.

### 3.2 MASKING FREQUENCY COMPONENTS

Not all the frequency components of all signals are important for the prediction of the final label. To filter these frequencies, we introduce a learnable mask for each frequency component of the signals. We use ReLU activations (Agarap, 2018) on the weights for the model to be able to mask the

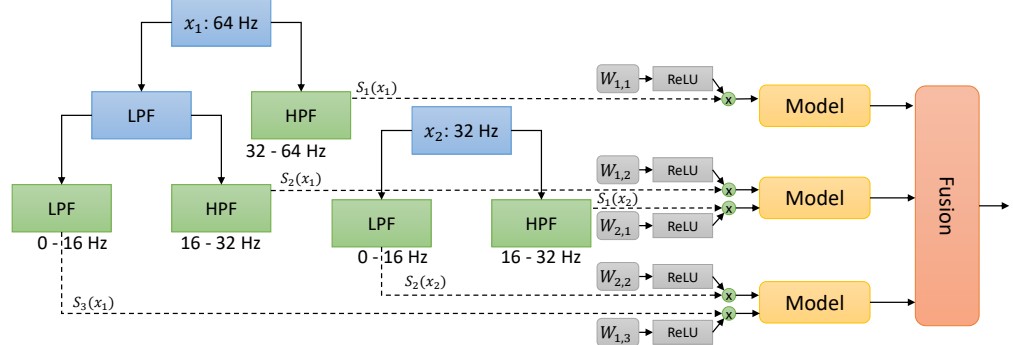

Figure 3: Structure of the model when there are two signal inputs; One that is sampled at 64Hz ($x_1$) and the other one at 32Hz ($x_2$). This figure shows how wavelet decomposition decomposes these signals into different frequencies with low pass filters (LPF) and high pass filters (HPF) and how we group these signals into different model components. This is a model-agnostic architecture so any model which works on multivariate time series data can be utilized.

uninformative components by setting the mask to zero so the input to component $i$ in the model is defined as:

$$I_i = \{ReLU(W_i^{(1)})S_i(x_1), ReLU(W_i^{(2)})S_i(x_2), ..., ReLU(W_i^{(m)})S_i(x_m)\}$$

In order to force the model to use a sparse mask for the feature components we add the $\ell_1$ norm of the weights to the final loss of the model:

$$Loss = \mathcal{L} + \alpha \ell_1(ReLU(W))$$

where $\mathcal{L}$ is the normal loss of the model which is usually defined by Mean-Squared-Error (MSE) for regression models and Cross-Entropy-Loss for classification models. $\alpha$ is a hyperparameter that determines the weight of the regularization term and sets a trade-off between minimizing the mask weights and the model's loss.

### 3.3 FINAL FUSION OF COMPONENTS

We tried many different fusion techniques, such as attention, mean, weighted average, ensemble methods, transformer fusion, as well as GradBlend (Wang et al., 2020b), Efficient Low-rank Multi-modal Fusion with Modality-Specific Factors (Liu et al., 2018), Du et al. (2018) but concatenating and passing through a fully connected layer worked best in our experiments. This is not surprising considering the unique structure of the inputs to the fusion layer, as each input will have some partial information about the target and should be combined to get the output. Note that MultiWave is agnostic to the choice of fusion technique, and while in our experiments concatenating and passing through a fully connected layer works best for other datasets, different fusion techniques can be utilized.

### 3.4 MODEL TRAINING

Algorithm 1 shows the MultiWave training procedure. We first train the model with feature masks included, and then remove the features that have a mask value of zero as they do not contribute to the model output anymore. We also remove all the components for which all the feature masks have values of zero. Then we add the baseline model with input of the original signals $X^{1:T} = \{x_1^{1:T}, x_2^{1:T}, ..., x_m^{1:T}\}$ as an additional component to the fusion model, and then train the model without the included masks. In this way, MultiWave can fall back to the baseline model (using early fusion of frequencies) if the addition of frequency components does not help, while also being able to use frequency components to improve the performance.

We used Weights & Biases Biewald (2020) for tracking and logging the experiments and Pytorch Paszke et al. (2019) to implement and train our models.

---

**Algorithm 1** MultiWave training procedure

---

Input $n$ samples with signals $X^{1:T} = \{x_1^{1:T}, x_2^{1:T}, ..., x_m^{1:T}\}$ with rates $R = \{r_1, r_2, ..., r_m\}$.
Apply wavelet decomposition and obtain $I_i = \{S_i(x_1), \ldots, S_i(x_m)\}$, for $i \in \{1 \ldots L\}$
Train fusion model with components $\Phi_i(I_i)$ and weights $W_i$ for $i \in \{1 \ldots L\}$, where each $W_i$
is made up of weights $W_i^{(j)}$, corresponding to subsignal $S_i(x_i)$, i.e. the part of the signal $x_j$ for
$j \in \{1 \ldots m\}$ provided to component $i$. The model is trained with ReLU masks, as shown in
Figure 3, which we denote $M(j)_i$, and which are 0 if the ReLU filtered out the corresponding
subsignal.
**for** Each component $\Phi_i$, $i \in \{1 \ldots L\}$ **do**
    **for** Each mask $M(j)_i$ **do**
        **if** $M(j)_i == 0$ **then**
            Set $W_i^{(j)} = 0$ (remove corresponding frequency)
        **end if**
    **end for**
    **if** $M(j)_i == 0$ for all $j$ **then**
        Set $\Phi_i = 0$ (remove component $i$)
    **end if**
**end for**
**if** AddBaseline is True **then**
    Add the baseline model with original signals as input $X^{1:T} = \{x_1^{1:T}, x_2^{1:T}, ..., x_m^{1:T}\}$ as an
additional component
    Continue to train the model without training the mask weights
**end if**

---

## 4 EXPERIMENTS

In this section, we evaluate the performance of MultiWave in synthetically generated and real-world
datasets.

### 4.1 SYNTHETICALLY GENERATED DATA

To determine the effectiveness of MultiWave in handling signals with different frequencies and
sampling rates, we generated synthetic data. The generated data consist of multiple square signals,
each with a different frequency and amplitude. The amplitude of each signal is randomly selected
from a uniform distribution ranging from 0-10. We then add uniform noise to the input data with
an amplitude of 3. The label $y$ is the sum of the amplitudes of the generated signals. The task is
to predict the label $y$ from given time series data. In Figure 4 results of 2 experiments are shown.
For the first experiment, we start with two square wave signals with frequencies of 1 and 2 Hz.
sampled at 128 Hz. for 1 second. Then we iteratively add two signals to the input with the same
sampling rate but frequencies of $\{4, 4, 8, 8, 16, 16, 32, 32\}$, respectively. The performance of the
LSTM and Transformer model with the addition of MultiWave are shown in the first column of
Figure 4. MultiWave delivers consistent and robust improvements with the addition of new signals.
In the second experiment, we generate two square-wave signals with frequencies of 2Hz and 4Hz
sampled, respectively, at 64 and 128 Hz., for 1 second. Then we reduced the sampling rate of the
first signal from 64 Hz. to 32, 16, 8, and 4 Hz. The results for the LSTM and Transformer models
with and without the addition of MultiWave are shown in the second column of Figure 4. Here again,
MultiWave consistently improves the performance of the baseline model.

### 4.2 THE WEARABLE STRESS AND AFFECT DETECTION (WESAD)

Wearable Stress and Affect Detection (WESAD) is a publicly available multimodal data set for
stress and affect detection. WESAD contains physiological response data from 15 subjects during
three sessions of baseline, amusement, and stress. The baseline session is 20 minutes, where the
subject is doing a neutral reading task, the amusement session is watching a set of funny videos for
392 seconds, and the stress session is when the subject is exposed to the Trier Social Stress Test
Kirschbaum et al. (1993) for 10 minutes. During these sessions, physiological measurements such as

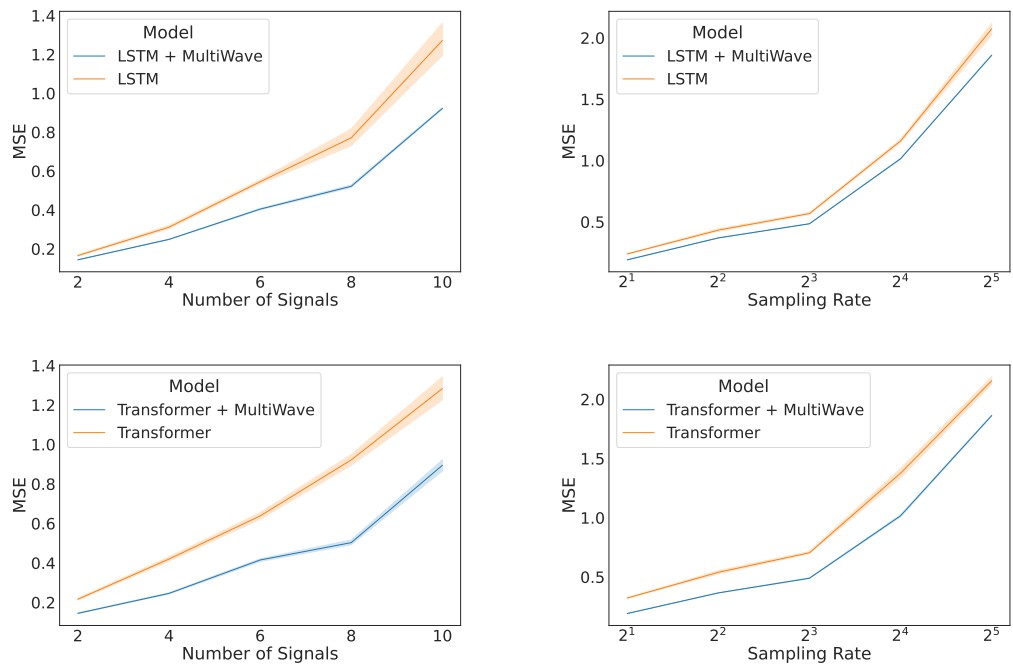

Figure 4: The Mean Squared Error (MSE) results on synthetic dataset, when we increase the number of signals (left column) and change the difference between the signal sampling rates (right column). First row describs the results for LSTM model while second row shows the results for Transformers.

| Dataset | Model | AUC without MultiWave | AUC with MultiWave |
|---|---|---|---|
| WESAD | LSTM | $0.822 \pm 0.04$ | $\mathbf{0.828 \pm 0.04}$ |
| | CNN-Attn | $0.831 \pm 0.03$ | $\mathbf{0.877 \pm 0.03}$ |
| | CNN-LSTM | $0.807 \pm 0.04$ | $\mathbf{0.839 \pm 0.04}$ |
| | FCN | $0.805 \pm 0.04$ | $\mathbf{0.833 \pm 0.05}$ |
| COVID-19 0 days ahead | LSTM | $0.983 \pm 0.008$ | $\mathbf{0.989 \pm 0.004}$ |
| | CNN-Attn | $0.978 \pm 0.012$ | $\mathbf{0.979 \pm 0.02}$ |
| | CNN-LSTM | $0.979 \pm 0.010$ | $\mathbf{0.981 \pm 0.012}$ |
| | Transformer | $0.980 \pm 0.007$ | $\mathbf{0.984 \pm 0.008}$ |
| COVID-19 12 days ahead | LSTM | $0.977 \pm 0.007$ | $\mathbf{0.979 \pm 0.006}$ |
| | CNN-Attn | $\mathbf{0.967 \pm 0.008}$ | $0.967 \pm 0.013$ |
| | CNN-LSTM | $0.961 \pm 0.012$ | $\mathbf{0.962 \pm 0.009}$ |
| | Transformer | $0.969 \pm 0.01$ | $\mathbf{0.972 \pm 0.01}$ |

Table 1: The AUC results on real-world datasets

blood volume pulse (BVP), electrocardiogram (ECG), electrodermal activity (EDA), electromyogram (EMG), respiration (RESP), temperature (TEMP), and accelerometer (ACC) are collected using a chest-worn device and a wrist-worn device. The chest-worn device collects data at 700 Hz. while the wrist-worn device collects data at 64, 32 and 4 Hz. We followed Dzieżyc et al. (2020) to preprocess the data. Unlike Dzieżyc et al. (2020), however, we use sampling rates that are powers of 2 to obtain more consistent signals. More details on the preprocessing of the signals in this data set are given in the Supplementary Section A.1.

| Frequency component | Features |
|---|---|
| $0 - \frac{1}{16}$ days | **High sensitivity C-reactive protein**, Glucose |
| $\frac{1}{16} - \frac{1}{8}$ days | **Lactate dehydrogenase** |
| $\frac{1}{8} - \frac{1}{4}$ days | D-D dimer |
| $\frac{1}{4} - \frac{1}{2}$ days | |
| $\frac{1}{2} - \frac{1}{1}$ days | **(%)lymphocyte**, **High sensitivity C-reactive protein** |

Table 2: The features with nonzero mask weights in COVID-19 dataset at the end of the training. These features were consistently selected in 5 different runs of the training procedure, indicating that they are informative for the predictive task, which is in concordance with domain expertise. MultiWave was able to automatically determine these dependencies, as well as determine which of the subsignals of different frequencies are relevant, as shown in Figure 5

The results achieved in this data set are shown in Table 1. Consistent with the results reported in Dzieżyc et al. (2020), CNN models achieve the best performance in this data set, and the Convolutional Neural Network with Attention (CNN-Attn) was presented as achieving the best performance in this data set [1]. We could not apply basic Transformers to this dataset, as the length of sequences leads to out-of-memory errors. MultiWave significantly improves the performance of all baseline models, since it allows the model components to learn short-term and long-term changes and can combine these multirate signals without the need for alignment and imputation. More details about these experiments are given in the Supplementary Section A.

## 4.3 COVID-19

The COVID-19 dataset Yan et al. (2020) is a publicly available data set that contains 74 indicators of 375 patient blood samples from 10 January to 18 February 2020 at Tongji Hospital, Wuhan, China. These indicators are collected in irregular time intervals and sampling rates range from 0 to 6 per day. We are interested in the task of predicting in-hospital mortality given the time series of biomarkers. The duration of hospital stays for patients varies from 2 hours to 35 days.

To process this data set, we sampled the features with different rates ranging from 1 to 8 day intervals (more details in the Supplementary Section B). If multiple values are recorded for a feature in the determined rate, we use the last recorded value. We fill in the missing values by linear interpolation. If a feature is completely missing for a patient, we use the mean value of that feature in all patients in the training set to fill these values. To evaluate the capability to select features in our model, we used all 74 features. Because of this, we were unable to use the original test set provided with the dataset as it only contains 3 features. Therefore, to evaluate our models, we separated 100 patients from the original data set and used 50 patients for validation and 50 patients to test our models.

We perform two groups of experiments first to predict mortality when all patient data leading to hospital discharge are included (0 days ahead prediction) and predicting mortality 12 days ahead of hospital discharge is included (12 days ahead prediction). The average AUC results for 5 runs of each experiment are included in Table 1. For this data set, we used the LSTM, CNN-Attn, CNN-LSTM, and Transformer models, and the results are reported with and without inclusion of the MultiWave framework. Consistent with the results reported in Sun et al. (2021), LSTM-based models achieve the best results, and MultiWave brings consistent improvements to all baseline models.

Yan et al. (2020) reports three features of lactic dehydrogenase, lymphocytes, and high-sensitivity C-reactive protein are the most important characteristics in the prediction of hospital mortality in this data set. To determine whether MultiWave can recognize the correct features in predicting the target, we looked at non-zero masked values in each model component for five different runs and we show the common ones in Table 2. As can be seen, these three features are consistently selected by model, and the important frequency of them is shown. In Figure 5 we show how the mask weights for these features across the components are changed over the training epochs of the model for one run.

---

[1]Note that the results are not directly comparable to this paper as we use different sampling rates and we use multi channel inputs to FCN rather than single channel used in this paper

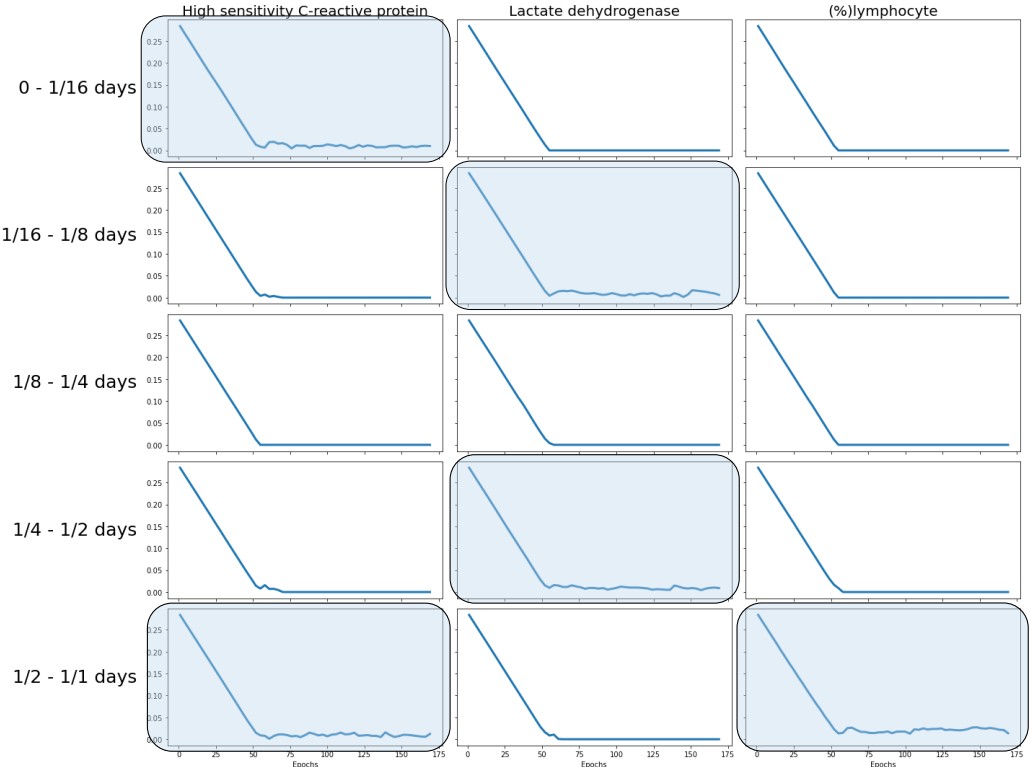

Figure 5: The mask weights over time for the three most important features for different frequency components in COVID-19 dataset. The masks that are not zeroed out are highlighted.

## 5 CONCLUSIONS

In this paper, we proposed a new framework called MultiWave that augments any deep learning time series model with components that operate at different frequencies of signals using wavelet decomposition. We further improved this model by introducing frequency masks, which remove non-informative frequency components of signals from the component inputs. We show that this framework improves the performance of time series models in synthetic datasets, as well as two real-world datasets for stress detection and COVID-19 prediction.

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

| Signal | Original Sampling | Downsampled to |
|---|---|---|
| ECG RespiBAN | 700 Hz | 64 Hz |
| ACC RespiBAN | 700 Hz | 8 Hz |
| EMG RespiBAN | 700 Hz | 8 Hz |
| EDA RespiBAN | 700 Hz | 4 Hz |
| TEMP RespiBAN | 700 Hz | 4 Hz |
| Respiration RespiBAN | 700 Hz | 4 Hz |
| BVP Empatica | 64 Hz | 64 Hz |
| ACC Empatica | 32 Hz | 8 Hz |
| EDA Empatica | 4 Hz | 4 Hz |
| TEMP Empatica | 4 Hz | 4 Hz |

Table 3: WESAD dataset feature frequencies

# A  WESAD

## A.1  PREPROCESSING

We follow Dzieżyc et al. (2020) to preprocess the data and use a similar technique to downsample the signals. However, instead of the frequencies used in the article, we use frequencies that are powers of 2 given in the table 3. We split data into five folds and for each fold we train the data on 3 folds, validate on 1 fold, and test on the last remaining fold. We do this 5 times and the mean and standard deviation are reported in the paper.

## A.2  HYPERPARAMETERS

We select hyperparameters for the baseline models based on the validation set using grid search for each model and select hyperparameters for the components of MultiWave such that the number of parameters is approximately equal between the two models. For other MultiWave hyperparameters, we select them using a grid search on the validation set as well. The hyperparameters selected for the WESAD data set are given in Table 6.

# B  COVID-19 DATASET

## B.1  PREPROCESSING

This is an irregularly sampled dataset, we resample features based on their overall rate in all training data according to Table 5. If a feature is missing in the time of sampling, we used the closest available value, if a feature is completely missing for a patient, we fill the values with average of that feature among all the training samples. We also normalize the features to values between 0 and 1 using a min-max scaler.

## B.2  HYPERPARAMETERS

We use the validation data to select the hyperparameters for the baseline model using grid search, and then for components of MultiWave, we set the model sizes such that the number of parameters is approximately the same between the two models. We use data from 50 subjects for validation. The hyperparameters selected for MultiWave in this dataset are given in Table 6.

| Model | Hyper parameter | Value |
|---|---|---|
| LSTM | LSTM cell size | 28 |
| | Initial Mask Weight values | 0.5 |
| | Mask norm weight | 0.05 |
| | Number of Layers | 1 |
| | Learning rate | 0.001 |
| | Batch size | 16 |
| | patience | 15 |
| | Number of Components | 6 |
| | AddBaseline | True |
| CNN-Attn | CNN Kernel size | 7 |
| | Initial Mask Weight values | 0.5 |
| | Mask norm weight | 0.1 |
| | Number of Layers | 2 |
| | Learning rate | 0.001 |
| | Batch size | 16 |
| | patience | 15 |
| | Number of Components | 6 |
| | AddBaseline | True |
| CNN-LSTM | CNN Kernel size | 7 |
| | Initial Mask Weight values | 0.5 |
| | Mask norm weight | 0.1 |
| | Number of Layers | 2 |
| | Learning rate | 0.001 |
| | Batch size | 16 |
| | patience | 15 |
| | Number of Components | 6 |
| | AddBaseline | False |
| FCN | CNN Kernel size | 7 |
| | Initial Mask Weight values | 0.5 |
| | Mask norm weight | 0.1 |
| | Number of Layers | 2 |
| | Learning rate | 0.001 |
| | Batch size | 16 |
| | patience | 15 |
| | AddBaseline | False |

Table 4: The selected MultiWave hyperparameters for WESAD dataset

| Signals | Sampled Rates |
|---|---|
| hemoglobin,Serum chloride,Prothrombin time,procalcitonin,eosinophils(%),Alkaline phosphatase,albumin,basophil(%),Total bilirubin,Platelet count,monocytes(%),indirect bilirubin,Red blood cell distribution width, neutrophils(%), total protein, Prothrombin activity, mean corpuscular volume, hematocrit, White blood cell count, mean corpuscular hemoglobin concentration, fibrinogen, Urea, lymphocyte count, Red blood cell count, Eosinophil count, Corrected calcium, Serum potassium, glucose, neutrophils count,D irect bilirubin, Mean platelet volume, RBC distribution width SD, Thrombin time, (%)lymphocyte, D-D dimer, Total cholesterol, aspartate aminotransferase, Uric acid, HCO3-,calcium,Lactate dehydrogenase, platelet large cell ratio, monocytes count, PLT distribution width, globulin,$\gamma$-glutamyl transpeptidase,International standard ratio,basophil count(#),mean corpuscular hemoglobin ,Activation of partial thromboplastin time,High sensitivity C-reactive protein,serum sodium,thrombocytocrit,glutamic-pyruvic transaminase,eGFR,creatinine | every day |
| antithrombin,Quantification of Treponema pallidum antibodies,HBsAg,HCV antibody quantification,Amino-terminal brain natriuretic peptide precursor(NT-proBNP),Fibrin degradation products,HIV antibody quantification,ESR | every 2 days |
| PH value | every 3 days |
| Interleukin 2 receptor,Interleukin 10,Interleukin 8,Tumor necrosis factor$\alpha$,Interleukin 1$\beta$,Interleukin 6 | every 5 days |
| 2019-nCoV nucleic acid detection | every 7 days |
| ferritin | every 8 days |

Table 5: COVID dataset feature frequencies

| Model | Hyper parameter | Value |
|---|---|---|
| LSTM | LSTM cell size | 19 |
| | Initial Mask Weight values | 0.3 |
| | Mask norm weight | 0.1 |
| | Number of Layers | 1 |
| | Learning rate | 0.001 |
| | Batch size | 16 |
| | patience | 25 |
| | Number of Components | 6 |
| | Reset Model | False |
| CNN-Attn | CNN Kernel size | 7 |
| | Initial Mask Weight values | 0.3 |
| | Mask norm weight | 0.1 |
| | Number of Layers | 2 |
| | Learning rate | 0.001 |
| | Batch size | 16 |
| | patience | 25 |
| | Number of Components | 6 |
| | Reset Model | False |
| CNN-LSTM | CNN Kernel size | 7 |
| | Initial Mask Weight values | 0.3 |
| | Mask norm weight | 0.1 |
| | Number of Layers | 2 |
| | Learning rate | 0.001 |
| | Batch size | 16 |
| | patience | 25 |
| | Number of Components | 6 |
| | Reset Model | False |
| Transformer | Hidden Embedding Size | 13 |
| | Initial Mask Weight values | 0.3 |
| | Mask norm weight | 0.1 |
| | Number of Layers | 4 |
| | Number of Heads | 3 |
| | Learning rate | 0.001 |
| | Batch size | 16 |
| | patience | 25 |
| | Reset Model | False |

Table 6: The selected MultiWave hyperparameters for COVID dataset

