# OpenReview forum: "MultiWave: Multiresolution Deep Architectures through Wavelet Decomposition for Multivariate Timeseries Forecasting and Prediction"
_ICLR.cc/2023/Conference — Submitted to ICLR 2023_

### Official Review · Reviewer_1QUg · 2022-10-21

**Confidence:** 3
**Correctness:** 3
**Technical Novelty And Significance:** 2
**Empirical Novelty And Significance:** 2
**Recommendation:** 3

**Clarity, Quality, Novelty And Reproducibility:**

- Overall the paper is clearly written.
- As mentioned above, the quality of the paper could be improved by adding more experiments that support the design decisions of the paper (e.g. evidence for: is the masking useful? How and for what?).
- while mentioned as a strength above, the fact that frequency decompositions are widely known, means the novelty of applying a pre-processing wavelet decomposition step is minimal (though it's interesting to do it in the form suggested).

- Some more information on the experiments would be interesting. E.g. in Fig 5: what's the distribution of masks across all features?

**Strength And Weaknesses:**

Strengths.
- the idea of the paper is clear and the paper can be nicely followed.
- frequency decomposition is a standard tool in signal processing, and hence combination with deep learning tools is promising
- learning the masking to prune uninformative frequencies is very promising

Weaknesses.
- Some parameter investigations that would provide insight into the method are missing: what is the impact of the mask sparsification parameter alpha on the performance? Is alpha=0 maybe the best case? Or where is the optimum? Also: The model likely performs best if the input signals are sampled at integer multiples - does performance suffer when this is not the case? By how much?
- Table 1 shows that the results are better, but only marginally so, hence I'd expect moderate impact (e.g. in the first row, I'd argue that it might not even be significant due to overlapping error-bars - if this is true, please bold all methods in these cases).

**Summary Of The Paper:**

The paper "MultiWave: Multiresolution Deep Architectures through Wavelet Decomposition for Multivariate Timeseries Forecasting and Prediction" combines wavelet decomposition with learnable time-series models. In addition to performing wavelet analysis to split the data into frequency components, a masking approach is proposed to select a sparse set of frequencies. The framework is shown to perform better on synthetic data, as well as two real-world datasets (WESAD & Covid19).


**Summary Of The Review:**

The paper is clearly written and can be nicely followed. However, the paper needs more work to carve out evidence for the fundamental design decisions (e.g. the masking method).

---

> ### Author Response · Authors · 2022-11-17
> **Thank you for your comments**
>
> Thank you for the helpful comments and detailed feedback! We really appreciate the time you took to consider our submission. We will consider all the reviews and resubmit an improved version of this work in the future. Wavelet decomposition is an extensively used tool in time series research but the novelty of this work is in the gating mechanism used and how the signals with different resolutions are combined through wavelet decomposition. In the future submission of this work, we will clarify and emphasize the contributions more clearly. To address the raised points:
> - For alpha, we are reporting the optimal value found for each dataset in the appendix, but we agree that an experiment showing how values of alpha affect the sparsity of the masks and the performance would be interesting to consider. We will include this in future versions of this work.
> - The goal of the table is to show in all cases there are some improvements (you would not lose performance by applying this method) but in the future, we will distinguish between the significant and non-significant improvements.

---

> > ### Comment · Reviewer_1QUg · 2022-11-18
> > **Thanks for the notes**
> >
> > Great! Good luck with the resubmission - looking forward to the paper!

---

### Official Review · Reviewer_6oFs · 2022-10-22

**Confidence:** 5
**Correctness:** 3
**Technical Novelty And Significance:** 2
**Empirical Novelty And Significance:** 2
**Recommendation:** 3

**Clarity, Quality, Novelty And Reproducibility:**

Can you provide an evaluation of the quality, clarity and originality of the work?
	The novelty is not enough.
	The writing quality should be improved.
	The reproducibility is simple and easy.


**Strength And Weaknesses:**

Strength. This paper proposes a framework that applies the wavelet transform to obtain different frequency time series signals and removes the irrelevant frequency for the given predictive task. Experiments have been evaluated on benchmark
deep learning framework and the benchmark data sets, which validates the effectiveness of the proposed methods.

Weaknesses.
	Applying the wavelet transform to analyse the time-series data set is not novel enough.
	The contributions should be emphasized.
	Besides, the novelty and the differences of this work should be stated in the text.
	Some description in the text should be improved. I.e. : and more recently transformers
	Why there is something missing about the features about the 1/4-2/1 days frequency components in the  Table 2.
	There are some errors about the punctuation and format should be corrected.
 I.e. ：Multirate time series classification;Frequency analysis of time series ...-->add the “:” or “.” behind the sentences.


**Summary Of The Paper:**

This paper proposed a framework, named MultiWave, which augments any deep learning time series model with components operating at the intrinsic frequencies of the signals. It applies discrete wavelet decomposition on each signal to obtain subsignals of different frequencies and groups all subsignals in the same frequency band together to train a component. Besides, it removes the irrelevant frequency components for the given predictive task. Experiments have been evaluated on three deep learning framework and three time series data sets.

**Summary Of The Review:**

This work proposed MultiWave, which augments any deep learning time series model with components operating at the intrinsic frequencies of the signals. It applies discrete wavelet decomposition on each signal to obtain subsignals of different frequencies and groups all subsignals in the same frequency band together to train a component. Besides, it removes the irrelevant frequency components for the given predictive task. However, it is not novel enough to apply to analyse time-series data sets. The contributions of this work should be emphasized.

---

> ### Author Response · Authors · 2022-11-17
> **Thank you for your comments**
>
> Thank you for the helpful comments and detailed feedback! We really appreciate the time you took to consider our submission. We will consider all the reviews and resubmit an improved version of this work in the future. Wavelet decomposition is an extensively used tool in time series research but the novelty of this work is in the gating mechanism used and how the signals with different resolutions are combined through wavelet decomposition. In the future submission of this work, we will clarify and emphasize the contributions more clearly. Thank you for pointing out ways to improve the text as well. Note that the 1/4-2/1 days frequency components are missing in Table 2 because there was no signal which consistently has non-zero masks for these frequencies. We will clarify this in a future version.

---

### Official Review · Reviewer_BvMP · 2022-10-22

**Confidence:** 3
**Correctness:** 3
**Technical Novelty And Significance:** 2
**Empirical Novelty And Significance:** 2
**Recommendation:** 3

**Clarity, Quality, Novelty And Reproducibility:**

Overall, the paper is well written and clear. Novelty needs further clarification as I mentioned in the weakness. Reproducibility looks pretty feasible.

**Strength And Weaknesses:**

Strength:
Using feature mask to learn the best Wavelet decomposition coefficients is interesting.

Weaknesses
My concern is on the novelty of the proposed method. The framework is close to the work as the authors referred: “Multilevel wavelet decomposition network for interpretable time series analysis”. Also experiment section could be improved.


**Summary Of The Paper:**

In this paper, the authors propose a new framework MultiWave for multivariate time series forecasting using Wavelet decomposition. Key contributions of the proposed MultiWave is that it can handle input signals with different frequencies and select most informative frequencies after the Wavelet decomposition using feature masks. Experimental results show efficacy of the proposed method on boosting existing time series forecasting methods.

**Summary Of The Review:**

1. For Algorithm 1, my understanding is that learning of the feature mask M is done in sequential iterations during the supervised learning of the forecasting task. All components with zero masks will be removed after the training. Please confirm the process above.
2. What happens if the training set contains samples with non-zero Wavelet decomposition coefficients at all levels? Feature mask will not kick in in this case and the proposed method will degenerate to feature extractor using the Wavelet decomposition.
3. Instead of oversampling signals with lower sampling rates, how about just concatenate all signals/features after the Wavelet decomposition and use them for the subsequent learning task? Another option is to pad signal with the low sampling rates with all zeros.
4. Experiments look a little bit weak. Can the proposed framework improve other state-of-the-art algorithms for time series forecasting? For example, deepar, N-beats, informer and state-space based forecasting models, to name a few.
5. One minor issue: in page 3, sec. 3.1, is it 30 HZ for signal x2?

---

> ### Author Response · Authors · 2022-11-17
> **Thank you for your comments**
>
> Thank you for the helpful comments and very detailed feedback! We really appreciate the time you took to consider our submission. We will consider all the reviews and resubmit an improved version of this work in the future.
>
> While wavelet decomposition is an extensively used tool in time series research, the novelty of this work is in the gating mechanism used and how the signals with different resolutions are combined through wavelet decomposition. To answer your questions:
>
> 1. Yes, that is true, but note that there is no difference in removing the components during the training or after it as if a component becomes zero, the gradient will be zero for that component so there is no way that it can become non-zero.
> 2. Yes, that is true, but in our experiments, this never happened, and usually, the wavelet coefficients are very sparse (most of them become zero after training) as the information needed for a task usually is in just one or two frequencies of the signals.
> 3. This can be done but this is not an optimal way of handling multi-rate signals as if they are padded the timings would not match between different signals and will lead to inferior performance. Concatenating will have similar problems, resulting in a loss of structure of the signals which is very important in these tasks.
> 4. Please note that in our experiments we focused on the problem of time-series prediction rather than forecasting and the datasets we used are in this field (have one label) so the methods you mentioned are not applicable here. We tried to compare against state-of-the-art methods for these datasets and tasks. In a future version of this work, we will incorporate forecasting tasks and will compare against SOTA in that area for those tasks.
> 5. Thank you for pointing this out, yes that is a mistake and it should be 30 Hz for x2.

---

### Official Review · Reviewer_jcQW · 2022-10-25

**Confidence:** 3
**Correctness:** 3
**Technical Novelty And Significance:** 2
**Empirical Novelty And Significance:** 1
**Recommendation:** 3

**Clarity, Quality, Novelty And Reproducibility:**

### Clarity:
The paper properly explains its approach. However the sequence of operations in Figure 3 is confusing. Is the ReLU function applied element-wise to the weights before the multiplication with the input? That does not seem right.

### Quality:
The experimental part presents promising initial results but currently does not attempt a comparison to established work. Without it the quality of the experimental section cannot be properly verified.
https://arxiv.org/pdf/2010.12493.pdf and https://www.mdpi.com/2227-7390/8/12/2125 study similar problems. A future version of this draft could also involve comparisons to other works. I am no expert here, I am merely using these two papers to back up the claim that similar work exists.

### Novelty:
Aspects of this work exists in prior work. I am not aware of another paper using precisely the presented approach. https://www.mdpi.com/2227-7390/8/12/2125 for example uses wavelet packets and fuses the different time scales via resampling.

### Reproducibility:
In the experimental sections, the results of multiple runs are reported. Do multiple runs use different seeds for initialization? If the exact seed values appeared in the paper, the results could perhaps be made reproducible.

**Strength And Weaknesses:**

Strengths:
- Fusing signals measured at different sampling rates is an important part of every time-series processing pipeline and therefore important.
- Using the wavelet transform for fusing purposes is an interesting approach.

Weaknesses:
- The experimental part makes no comparisons to existing work in related papers.
- Time-series processing is a long-established field.
   - See: i.e. https://arxiv.org/pdf/2010.12493.pdf for a review which also studied the covid-19 data set.
   - Or https://www.mdpi.com/2227-7390/8/12/2125 for a paper which is also using wavelets to process medical time-series data.
- Comparing the numbers from this study to the results from other papers is an important part of the process of assessing if this work is suitable for ICLR. I am not claiming that comparison to precisely the two papers above is necessary, but a future version of this draft should include leading papers in the field and compare to them.


**Summary Of The Paper:**

The paper "MULTIWAVE: MULTIRESOLUTION DEEP ARCHITECTURES THROUGH WAVELET DECOMPOSITION FOR MULTIVARIATE TIME SERIES FORECASTING AND PREDICTION" proposes to fuse time series data with different sampling rates via concatenation of transformed features at matching scales in the wavelet domain.
After motivating and explaining the approach the paper proceeds to test their approach on the "Wearable Stress and Affect Detection" dataset and a COVID-19 mortality prediction problem.



**Summary Of The Review:**

This paper is not properly embedded into the existing literature in its current form. Unfortunately, I am no expert in the field of medical time series processing, so I can't suggest the leading papers in the field for comparison. However, I am confident in my assessment, that no comparison to existing work is not enough.

---

> ### Author Response · Authors · 2022-11-17
> **Thank you for your comments**
>
> Thank you for the helpful comments and very detailed feedback! We really appreciate the time you took to consider our submission. We will consider all the reviews and resubmit an improved version of this work in the future.
>
> While wavelet decomposition is an extensively used tool in time series research, the key novelty of this work is in the gating mechanism used and how the signals with different resolutions are combined through wavelet decomposition. We will make this clearer in the next version of the paper.
>
> For the experiments, our tool is not designed to handle irregularly sampled time-series data and is instead designed for multi-resolution data, so we compared our model to the models that work best for such signals and tried to compare with the best such models in the literature. In the future submission, we will compare the COVID-19 dataset performance against models that work specifically on irregularly sampled time-series data as well.
>
> Notably, our models *do* improve the state-of-the-art on the WESAD dataset, as the previous state-of-the-art in deep learning models was achieved using CNN-LSTM and CNN-Attn, which we compared against and which MultiWave outperformed.

---

> > ### Comment · Reviewer_jcQW · 2022-11-24
> > **Please cite specific numbers.**
> >
> > Thank you for considering the review. Please consider directly citing the numbers from related approaches on the WASAD-Dataset in Table 1. Such an addition will help assure your readers of the good quality of your work.

---

### Decision · Program_Chairs · 2023-01-20

**Decision:**

Reject

**Justification For Why Not Higher Score:**

There is a consensus that the paper is ready for publication. This apparently shared by the authors.

**Justification For Why Not Lower Score:**

N/A

**Metareview: Summary, Strengths And Weaknesses:**

The reviewers expressed concerns about the novelty of the method and the work was not positioned adequately wrt the state of the art. I would encourage the authors to better motivate and compare their work with related work, both, from a methodological perspective as from an experimental evaluation perspective.

**Summary Of Ac-Reviewer Meeting:**

N/A